# Understanding Information Storage and Transfer in Multi-modal Large Language Models

**Samyadeep Basu**
University of Maryland

**Martin Grayson**
Microsoft Research

**Cecily Morrison**
Microsoft Research

**Besmira Nushi**
Microsoft Research

**Soheil Feizi**
University of Maryland

**Daniela Massiceti**
Microsoft Research

## Abstract

Understanding the mechanisms of information storage and transfer in Transformer-based models is important for driving model understanding progress. Recent work has studied these mechanisms for Large Language Models (LLMs), revealing insights on how information is stored in a model's parameters and how information flows to and from these parameters in response to specific prompts. However, these studies have not yet been extended to Multi-modal Large Language Models (MLLMs). Given their expanding capabilities and real-world use, we start by studying one aspect of these models – how MLLMs process information in a factual visual question answering task. We use a constraint-based formulation which views a visual question as having a set of visual or textual constraints that the model's generated answer must satisfy to be correct (e.g. What movie directed by *the director in this photo* has won a *Golden Globe*?). Under this setting, we contribute i) a method that extends causal information tracing from pure language to the multi-modal setting, and ii) *VQA-Constraints*, a test-bed of 9.7K visual questions annotated with constraints. We use these tools to study two open-source MLLMs, LLaVa and multi-modal Phi-2. Our key findings show that these MLLMs rely on MLP and self-attention blocks in much earlier layers for information storage, compared to LLMs whose mid-layer MLPs are more important. We also show that a consistent small subset of visual tokens output by the vision encoder are responsible for transferring information from the image to these causal blocks. We validate these mechanisms by introducing MULTEDIT, a model-editing algorithm that can correct errors and insert new long-tailed information into MLLMs by targeting these causal blocks.

## 1 Introduction

Multi-modal Large Language Models (MLLMs) trained on both text and images are rapidly moving from research into deployment and are being used by millions of people. Yet, while there have been some advances in understanding how Large Language Models (LLMs) work, much less has been done to understand MLLMs. This paper begins to close this gap by studying how information is stored and transferred in MLLMs. To do this through the lens of a factual Visual Question Answering (VQA) task – a very common use case of MLLMs today [26, 35, 27].

MLLMs process factual information in two steps: information storage and information transfer. Information storage refers to how facts from a pre-training dataset are stored in a model's parameters – its so-called 'parametric' memory. Information transfer describes how information from input prompts are propagated through these storage locations to the model's final output. Understanding

38th Conference on Neural Information Processing Systems (NeurIPS 2024)., Vancouver.

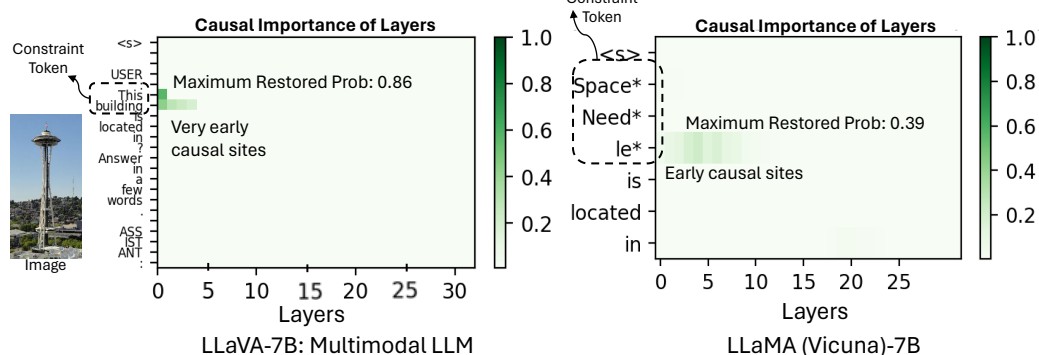

Figure 1: **MLLMs retrieve information from earlier internal layers compared to their LLM counterparts.** We find that very early MLP layers [1-4] have high indirect estimation effects to outputs (i.e., they are causal) in LLaVa-7B, whereas the middle MLP layers [4-7] are causal in LLaMA (Vicuna)-7B. For LLaMA, a larger window size (e.g., 5) is also required to find causal sites, compared to a window size of 1 for LLaVA-7B.

these mechanisms can have many benefits, including ensuring models are factually grounded and informing better evaluation protocols.

Extensive works have explored how LLMs store and transfer factual information [12, 29, 24], however, this has not been studied for multi-modal inputs. For example, it is suggested that auto-regressive Transformer-based LLMs store factual information in their mid-layer MLP parameters [12, 24]. However, MLLMs and LLMs are different: an MLLM involves an additional (continuous) image input, alongside a (discrete) text prompt, and requires additional modules to process it [26, 18, 19, 16, 7]. Typically, a vision encoder (e.g. CLIP) is used to convert this image into either visual tokens via a projection layer [18, 19, 11] or cross-attention layers [3, 7, 16, 36] which are then integrated into the language encoder. These differences suggest that our existing understanding of LLM information storage and transfer may not map directly to MLLMs.

In this work, we use a factual VQA task to study the mechanisms of multi-modal information storage and transfer. We use a constraint-based formulation which views a visual question as having a set of either visual or textual constraints (e.g. What movie directed by *this director* has won a *Golden Globe*?). The information retrieved by the model should satisfy these constraints (e.g. *this director*, *Golden Globe*) for the answer to be factually correct. This formulation, therefore, offers a systematic way of understanding a model's behavior. Under this framework, we propose MULTI-MODALCAUSALTRACE, which extends LLM causal tracing [12, 28, 33] to the multi-modal setting, to understand information storage, as well as leverage attention contribution methodologies [37, 8] to study information transfer in MLLMs. We also introduce *VQA-Constraints*, a new dataset of 9.7k factual questions annotated with constraints, spanning natural images (from OK-VQA [22], WikiMovies [37], and Known [12]). With these tools, we study how a widely-used MLLM family processes multi-modal information, specifically LLaVa [18, 19] and multi-modal Phi-2 [11][1].

Our key findings are that MLLMs, in contrast to LLMs: 1) retrieve information from earlier MLP layers (i.e. layers 1-4 vs layers 4-7 in a LLM) (see Fig. 1); and 2) use less parametric memory (require a smaller window size to retrieve this information), when answering a multi-modal question. We also find that information is transferred from a given image to these early MLP blocks through a consistent subset of visual tokens (e.g. last ∼36 tokens from LLaVa's CLIP encoder), and that the self-attention blocks in the middle layers are primarily responsible for shuttling this information to the last token.

Finally, we demonstrate that by editing these early causal MLPs, we can correct errors and insert new factual information into an MLLM. Specifically, we propose a new model-editing algorithm, MULTEDIT, which modifies the projection matrix of the early causal MLPs with a closed-form update. We empirically show MULTEDIT's effectiveness on questions from *VQA-Constraints* and Encyclopedic VQA [25].

In summary, our contributions are:

---

[1]Taken from the Bunny repository.

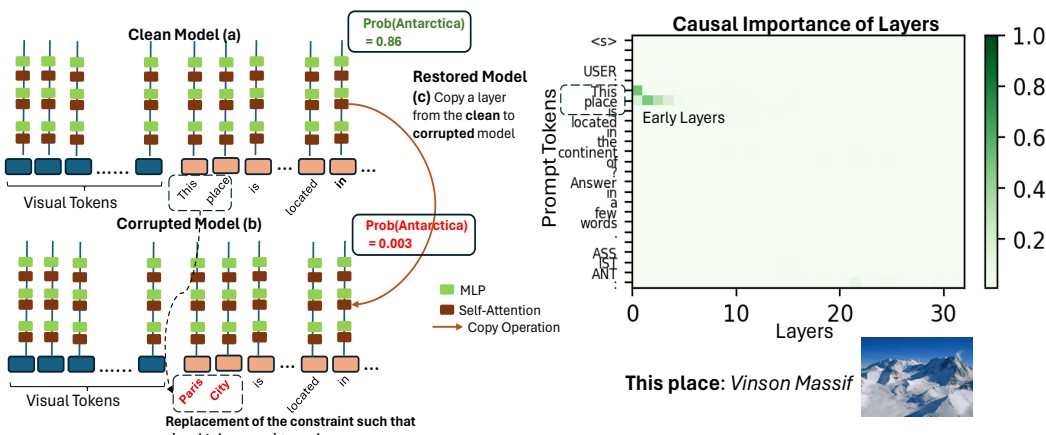

**Figure 2:** **We introduce MULTIMODALCAUSALTRACE, a causal tracing method to understand information storage in MLLMs.** A clean model is corrupted by replacing the question's constraint with an incorrect one for the given image (e.g. "This place" –> "Paris city" for an image of "Vinson Massif"). The activations of windows of layers are then iteratively copied from the clean to the corrupted model until the corrupted model restores its output probability to match the clean model's.

1. A novel multi-modal causal tracing methodology that can be used to study information storage from image and text inputs in MLLMs.
2. A new dataset, *VQA-Constraints*, of 9.7K factual visual questions about natural images annotated with constraints to support future research in these directions.
3. A suite of novel insights on the mechanisms underlying multi-modal information storage and retrieval in MLLMs.
4. A model-editing method, MULTEDIT, which we demonstrate can precisely correct erroneous information and insert new long-tailed information in an MLLM.

## 2   Related Works

**Multimodal Large Language Models.** We consider a MLLM to be a model that takes an image and text as input, and generates a text output [2]. Over the last year, such models have made tremendous advances in tasks like VQA and image captioning, including BLIP [15], BLIP-2 [16], Instruct-BLIP [7], LLaVA [18, 19], Flamingo [3] and multi-modal Phi-2 (from the Bunny repo) [11]. These MLLMs can broadly be categorized into two families based on how their visual information is integrated into the language model: (i) by embedding the vision encoder's output into each layer of the language model with a cross-attention layer (e.g., Flamingo, BLIP) or, (ii) by mapping the vision encoder's output into "visual tokens" in the language model's input space (i.e. alongside the text tokens) via a projection layer (e.g., LLaVA, Bunny). Both families are widely used, however, the projection layer family has recently shown stronger performance on popular benchmark [18, 19, 11]. We, therefore, focus our study of information storage and transfer on this model family.

**Interpretability of MLLMs.** A well-established arm of model interpretability examines the relationship between a model's performance and its internals. A range of recent works have studied the internal mechanisms of information storage [23, 34, 24] and transfer [9, 37] in LLMs. However, to the best of our knowledge, only a few works [30, 31] have studied the interpretability of MLLMs, with none specifically investigating the relationship between a model's outputs and its internal states. [30], for example, designs an interactive interface to visualize the attention maps in an MLLM, while [31] explores the shortcomings of the CLIP vision encoder in MLLMs. Neither consider the influence of both vision *and* text inputs on model internals or offer causal insights, as our work does. Our model editing approach which targets the projection layer MLLM family, is complemented by [6], who propose baselines for inserting information into the cross-attention layer MLLM family.

## 3   A Constraint-Based Framework for Studying Information Storage and Transfer in MLLMs

In this section, we describe the constraint-based formulation we use to study information storage and transfer in MLLMs. Under this framing, we introduce MULTIMODALCAUSALTRACE, a novel causal information tracing technique which we use to study multi-modal information storage. We also describe how we use attention contributions [8] to study multi-modal information transfer. Finally, we describe *VQA-Constraints*, a new test-bed of visual questions annotated with constraints.

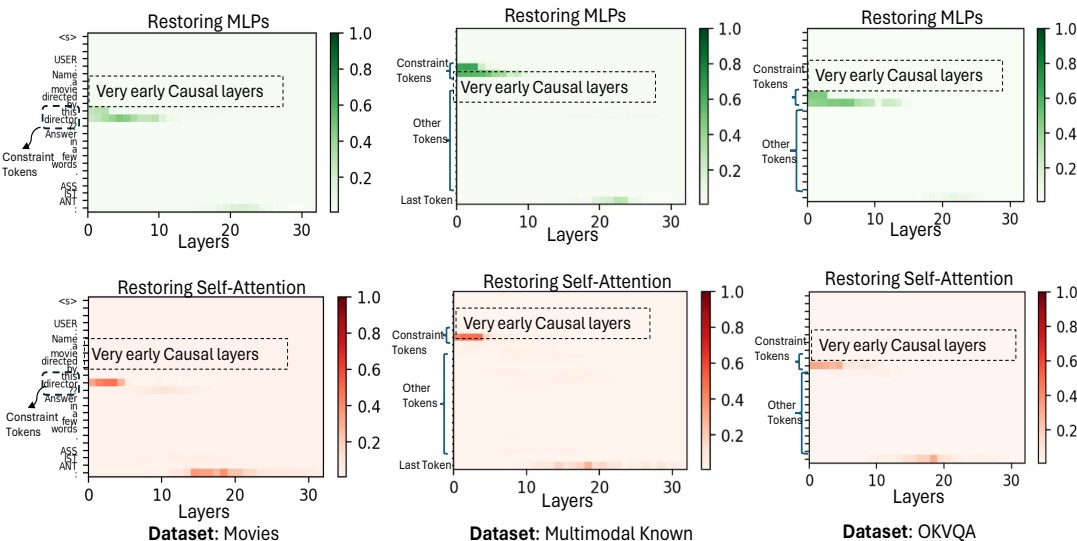

Figure 3: **Information to answer a visual question with a single constraint is mainly retrieved from early MLP and self-attention layers in MLLMs.** MULTIMODALCAUSALTRACE obtains high *indirect estimation effect* values in LLaVa's early MLP and self-attention blocks corresponding to the visual constraint, across all 3 datasets in *VQA-Constraints*. This suggests these layers are causally important for information storage. The causal traces emerge with a window size of 3 (see results with a window size of 1 in Appendix C).

## 3.1 A Multi-modal Constraint-based Framework

Prior works have used a constraint satisfaction framework to study LLMs [37, 20, 14] as they offer a systematic way of studying model behavior. This framing defines a constraint as a set of words in the question. The model must retrieve information that satisfies this constraint from its parametric memory in order to generate the correct answer. For example, for the question "What city is the *Space Needle* in?", the model must retrieve information relevant to the constraint "Space Needle". In the multi-modal setting, we consider these textual constraints as well as introduce visual constraints. We define a visual constraint to be a set of words in the question which refers to an entity in the image. The model must similarly retrieve information about this entity to generate the correct answer. For example, given an image of Christopher Nolan and the question "Name a movie directed by *this director* in *2006*?", the visual constraint is "*this director*" (and the text constraint is "*2006*"). We refer to a question involving both a visual and text constraint as a multi-constraint question, and ones with only a visual constraint as a single-constraint question.

We use this framework to study the widely-used "projection layer" MLLMs family. These models are composed of a visual encoder $f_\theta$, a large language model $g_\phi$ and a projection head $p_\gamma$. The projection head $p_\gamma$ is responsible for mapping the output of the visual encoder into the input space of the language model, as so-called "visual tokens". Given an image-text pair denoted as $(\mathbf{x}, y)$, the language model processes them as $g_\phi(p_\gamma(f_\theta(\mathbf{x})), h(t(y)))$, where $p_\gamma(f_\theta(\mathbf{x})) = \{\mathbf{v}_i\}_{i=1}^N$ is the set of visual token embeddings and $t(y) = \{t_i\}_{i=1}^M$ is the tokenized text inputs for the language model. These text tokens are processed by an embedding layer $h$ to obtain text token embeddings as $\{e_i\}_{i=1}^M \in \mathbb{R}^d$. Because we are interested in studying the outputs of specific layers in response to specific tokens, we use $g_\phi(.)_{k,\ell}$ to denote the output layer embedding corresponding to the $k^{th}$ token position and the layer $\ell$. We denote the output of a MLP layer as $g_\phi(.)_{k,\ell_{mlp}}$ and the output of a self-attention block as $g_\phi(.)_{k,\ell_{attn}}$.

## 3.2 MULTIMODALCAUSALTRACE: Studying Information Storage in MLLMs

Causal tracing, derived from the causality literature [28], can be combined with a constraint-based framework to gain causal insights on how information propagates through a model with respect to specific constraint tokens. This has been used to identify where information is stored in LLMs [23, 24] and text-to-image generative models [4]. The central idea of causal tracing is to corrupt a clean model by perturbing the input prompt. The activations of a small subset of layers are then iteratively copied

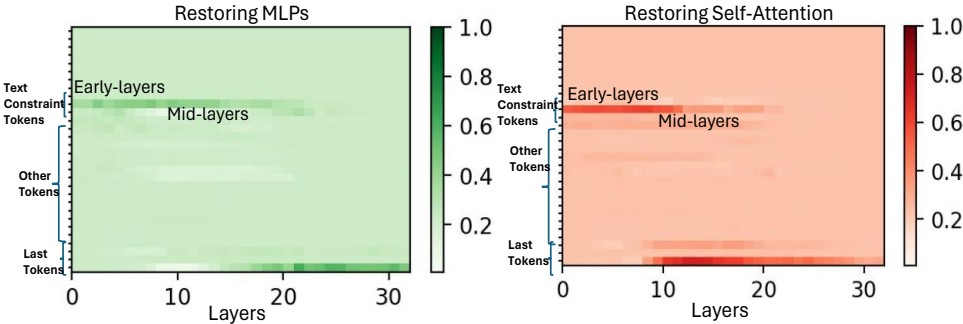

Figure 4: **Information to answer a visual question with a visual and textual constraint is retrieved from early *and* middle MLP and self-attention layers in MLLMs. This suggests that meeting multiple constraint requires more parametric memory compared to single constraints.** We show that MULTIMODAL-CAUSALTRACE obtains high indirect estimation effect values in the early and middle layers in LLaVa's on the OK-VQA dataset in *VQA-Constraints* (see multi-constraint results from the Movies dataset in Appendix F).

from the clean model to the corrupted model, until the corrupted model restores its output probability to match the clean model's. In this way, we can identify which layers are used to retrieve information relevant to the constraints in the prompt (i.e. causally relate the input to the output).

In LLMs, the model is corrupted by adding a small amount of Gaussian noise to the embeddings of the textual constraint tokens (usually less than 5) [23]. In MLLMs, however, noise must be added to a much larger number of token embeddings – those of the visual tokens (e.g., 576 in LLaVA and 729 in multi-modal Phi-2) from the projection head, *and* the textual tokens of the visual constraint (e.g. "*this director*"). Our experiments show that this large noise injection makes it difficult to revert the MLLM to a clean state and recover relevant causal traces (see Fig. 7).

We, therefore, introduce MULTIMODALCAUSALTRACE (see Fig. 2) to address this. Rather than adding Gaussian noise to the embeddings, we instead corrupt the visual constraint token IDs by replacing them with token IDs from a separate word or phrase, such that the visual information is ignored. We illustrate this through an example. **(1) Clean Model**: Given an image **x** of the Space Needle, and the question $y$, "Which city is *this building* located in?", we compute the probability of the model's output $O$ (e.g., "Seattle") as $\mathcal{P}_{clean}(O)$. **(2) Corrupted Model:** We substitute the visual constraint with an alternative such that the question does not require information from the image to be answered (e.g., "this building" is replaced with "Taj Mahal"). For a multi-constraint question which also has a textual constraint, we can either add Gaussian noise to the textual constraint tokens' embeddings (since there are only a few) or we can similarly replace the token IDs. After all replacements, we ensure that the question still makes semantic sense. We then measure the probability of original output $O$ as $\mathcal{P}_{corr}(O)$. With the right corruption, $\mathcal{P}_{corr}(O)$ is expected to be low. **(3) Restored Model:** We then iteratively copy layer activations $g_\phi(.)_{k,\ell_{mlp}}$ and $g_\phi(.)_{k,\ell_{attn}}$ $\forall k \in [1, M+N]$ from the clean model to the corrupted model, for each layer in turn. For a given layer $\ell$ and token position $k$, we denote the restored probability as $\mathcal{P}_{restored}(O)_{k,\ell}$. After a layer is copied, we observe if $\mathcal{P}_{restored}(O)_{k,\ell}$ is high – indicating that layer $\ell$ has a strong causal association to the output $O$. In some cases, no layers have a causal association. We note that this copy operation can be performed over a window of layers $\{\ell_i\}_{i=1}^{W}$ at a time, where $W$ is the window size. A window size of 1 copies only one layer at a time. Similar to [23], we track the *indirect estimation effect* for a layer $\ell$ as $\mathcal{P}_{restored}(O)_{k,\ell} - \mathcal{P}_{corr}(O)$ and use it as a metric to track causal states. Intuitively, this measures the difference in the probability of $O$ under the corrupted model and when a layer $\ell$ is restored to its original clean state. A high value indicates that the copied layer/s can restore the model to its original clean state (i.e. the layer is causal).

### 3.3 Studying Information Transfer in MLLMs with Attention Contributions

MULTIMODALCAUSALTRACE enables us to identify the specific layers a model retrieves information from in order to answering a visual question. A second component of understanding how MLLMs process factual information is understanding how input prompts are propagated through these storage locations to the model's final output. For this, we use attention contributions [8, 37] which compute how much one set of input tokens influences a set of output tokens during the self-attention operation.

Specifically, we use this to track (i) how information is transferred from the visual tokens to the causal layers, and (ii) from these layers to the final token, where the final probabilities are computed.

**Defining Attention Contributions.** The attention operation in a Transformer [32] consists of the query, value, key and output weight matrices: $W_q^\ell, W_v^\ell, W_k^\ell, W_o^\ell$. Each of these are divided into $H$ heads as $W_q^{\ell,h}, W_v^{\ell,h}, W_k^{\ell,h} \in \mathbb{R}^{d \times d_h}, W_o^{\ell,h} \in \mathbb{R}^{d_h \times d}$, where $d$ is the dimension of the internal token embeddings and $d_h$ is the dimensionality of the token embedding for a particular attention head $h$. We define the attention contribution from a token $j$ to token $i$ in layer $\ell$ as follows:

$$a_{ij}^\ell = \sum_{h=1}^{H} A_{ij}^{l,h} (g_\phi(.)_{j,\ell-1} W_v^{\ell,h}) W_o^{\ell,h} \tag{1}$$

where the attention matrix for layer $\ell$ and head $h$ is defined as:

$$A^{\ell,h} = softmax\left(\frac{(g_\phi(.)_{1:M+N,\ell-1} W_q^{\ell,h})(g_\phi(.)_{1:M+N,\ell-1} W_k^{\ell,h})^T}{\sqrt{d/H}}\right) \tag{2}$$

where $g_\phi(.)_{1:(M+N),\ell-1} \in \mathbb{R}^{(M+N) \times d}$ and $A^{\ell,h} \in \mathbb{R}^{(M+N) \times (M+N)}$. For understanding information transfer from the visual tokens to the causal layers, we use $j \in [1, M]$ and $i = c_{\text{constraint}}$, where $c_{\text{constraint}}$ corresponds to the last token in the visual constraint. For understanding the information transfer to the last token, we set $j = c_{\text{constraint}}$ and $i = c_{\text{last}}$, where $c_{\text{constraint}}$ corresponds to visual constraint's last token and $c_{\text{last}}$ corresponds to the last token in the question.

### 3.4 *VQA-Constraints*: A Constraint Annotated Test-Bed for VQA

Alongside the above tools, we also introduce a new test-bed called *VQA-Constraints* to enable our analyses. The test-bed consists of $9.7K$ natural images paired with factual questions, where each question is annotated with visual and textual constraints (see Sec. 3.1). Specifically, we source image-question pairs from the following datasets i) OK-VQA [21] which covers general knowledge questions, ii) Movies [37] which includes questions about movie directors and awards from Wikipedia, and iii) Known [23] which covers questions about countries, famous people, and places. The questions from the Movies and Known datasets are not originally multi-modal, so we modify them to refer to images which we source from Bing. We provide more details on the dataset construction and the constraint annotation in the Appendix B.

We leverage GPT-4 [27] to annotate the textual and visual constraints in the visual questions in *VQA-Constraints*. We first manually annotate 100 examples from each dataset with constraints. We provide this as context to GPT-4 and prompt it (see Appendix B) to annotate the constraints in new questions from Multimodal Known and OK-VQA. Due to the templated prompts in Multimodal Movies (e.g., "Name a movie directed by *this director*? "), the constraints are constant ("*this director*") across all examples and does not require external annotations.

## 4 Key Findings in how MLLMs Store and Transfer Information

Using the tools presented in Sec. 3.1, we present our key findings in how MLLMs retrieve information from internal layers and how this information is transferred across the model.

### 4.1 Finding 1: Early MLPs and self-attention layers are causal

We find that information required to answer a visual question is mainly retrieved from the early-layer MLP and self-attention blocks of a MLLM. This is confirmed by the high indirect estimation effects which MULTIMODALCAUSALTRACE assigns to the early layers in both LLaVa (see Fig. 3, Fig. 8) and multi-modal Phi-2 (see Appendix E) across the three datasets in *VQA-Constraints*.

This contrasts earlier results for LLMs which have been shown to retrieve information from mid-layer MLPs to answer factual questions [23, 24]. To obtain a fairer comparison, we apply MULTIMODAL-CAUSALTRACE to LLaMA and LLaVa which use the same language backbone. We run both on the same set of questions – LLaMA on the Known dataset [23], and LLaVa on our multi-modal version of Known which modifies the questions to refer to an image (see Sec. 3.4). In Fig. 1, we show that the MLPs in the first 4 layers are causal for LLaVa while the MLPs in layers 4-7 are causal for LLaMa.

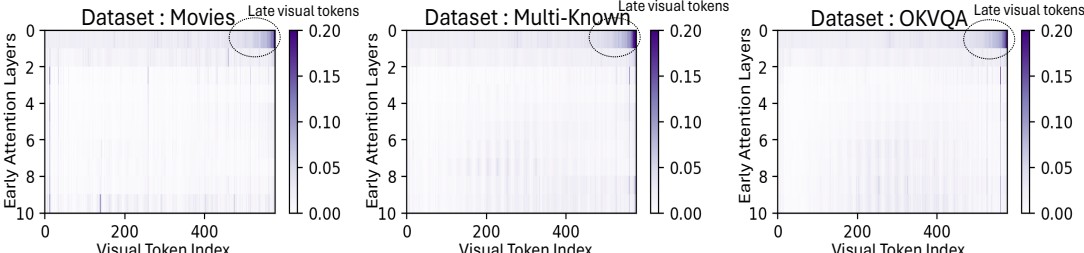

Figure 5: **The late visual tokens are primarily responsible for transferring information from the image to the early causal layers, via the first self-attention layer.** We visualize attention contributions (see Eq.(1)) from the visual tokens to the visual constraint token averaged across the three datasets in *VQA-Constraints*.

We also find that causal traces can be extracted from LLaVA with a minimum window size of 1, while LLaMa requires a minimum window size of 5 to obtain any significant causal traces.

Although we find a smaller window size of 1 to provide relevant causal traces (see Fig. 8), we find that a slightly larger window size of 3 results in consistent causal traces across all the questions. In Fig. 3, we report the results using a window size of 3, where we find that the early MLPs as well as self-attention layers are used to retrieve relevant knowledge to answer a visual question. We note that this observation is consistent for all the datasets in *VQA-Constraints*.

For multi-constraint questions that consist of a visual and textual constraint, information corresponding to the textual constraint is retrieved from a broader set of layers – *both* the early and mid-layer MLP and self-attention blocks (see Fig. 4). We also find that a larger window size (at least of 6) is required to obtain any causal traces. This suggests that more parametric memory is required to meet both a visual and textual constraint in a given question.

### 4.2 Finding 2: Only a subset of visual tokens are involved in transferring information from the image to the early causal MLP layers.

We also find that the late visual tokens transfer information from the image to the early causal MLP layers (where information is mainly stored) via the first self-attention layer. We show this in Fig. 5 by visualizing the attention values in the early self-attention layers between each visual tokens and the final token in the visual constraint using the method described in Sec. 3.3. We see that the values are the highest in LLaVa's first self-attention layer (which occurs just before the first causal MLP layer), specifically for the last subset of visual tokens (indexes 540-576 out of 576 in total). We hypothesize that these tokens may be summarizing image information that is relevant to the given question before it is transferred and then stored in the MLPs, however we leave this to future study. We note that this pattern holds across all three datasets from *VQA-Constraints*.

### 4.3 Finding 3: Mid-layer self-attention layers are involved in transferring information from the early causal layers to the question's final token

Rather counter-intuitively, we find that even though information is stored in the early layers, the self-attention blocks in the middle layers (rather than layers immediately after) are responsible for propagating this information to the question's final token. The model's answer is sampled at this point, hence the information present here likely influences the ultimate generation. We see this in Fig. 15 and Fig. 16, which plots the attention contribution values between the last visual constraint token and the last token in the question across all the layers, using the methodology in Sec. 3.3. For LLaVa, the self-attention blocks in layers 16-17 are most active. This behaviour is similar in LLMs which also use mid-layer self-attention blocks to transfer information from (mid-layer) stored locations to the last token position.

### 4.4 Finding 4: Mid-layer self-attention contributions can be used to predict whether a MLLM will generate a correct answer, but model confidence is a more reliable predictor

When a MLLM generates a correct answer, we observe that the self-attention contributions in its middle layers are higher to when it generates an incorrect answer (see Fig. 15). For LLaVa, this is specifically the attention contributions between the last constraint token and the last prompt token in the 16th and 17th layer. This holds potential to detect when a model will answer correctly without running a full inference pass, therefore enabling "early" failure mode detection. To inves-

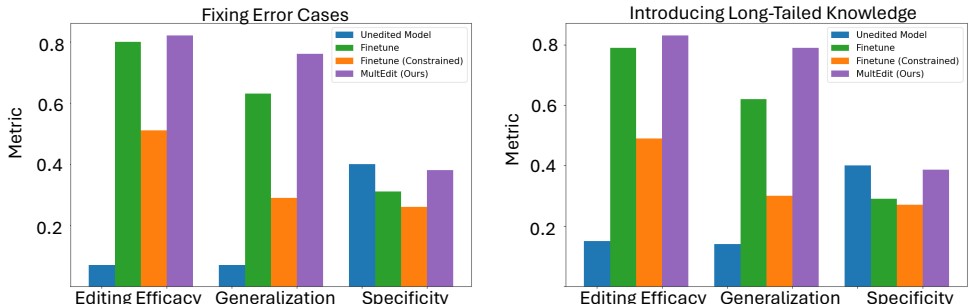

Figure 6: **MULTEDIT can correct error cases (left) and insert long-tailed factual knowledge (right) by editing the early causal MLP layers in an MLLM**. We use the average probability of the correct token $O^*$ as the metric for *Editing Efficacy* and *Generalization*, and VQA-Accuracy for *Specificity* (higher the better for all).

tigate this, we compute the scalar average of the attention contributions from these two layers as $\frac{a_{c_{\text{last}}, c_{\text{constraint}}}^{16} + a_{c_{\text{last}}, c_{\text{constraint}}}^{17}}{2}$ and find that it can classify a correctly generated answer with an AUROC of 0.63 on the Known dataset in *VQA-Constraints* (see Fig. 16). We find, however, that the model's confidence – the probability of the generated output token at the final layer – is a slightly stronger predictor, with an AUROC of 0.76 (see Fig. 17). This parallel's previous work in LLMs [37] which used the attention contributions from *all* the LLM's layers (via a linear model) to predict the generated answer's correctness. In comparison, our results suggest that a subset of MLLM's middle layers alone can be leveraged as a coarse "early" failure mode detector.

> **Takeaway.** MLLMs behave differently in terms of information retrieval from their parametric memory however quite similarly to LLMs in terms of information transfer to the final token.

## 5 Correcting and Inserting Long-Tailed Information in MLLMs

Previous works have shown that counterfactual information can be inserted into LLMs by editing their causal layers [23, 24]. In this section, we verify if a similar approach can be used to edit a MLLM. Specifically, we introduce MULTEDIT, which applies a closed-form update to the early causal MLP layers we identified in Sec. 4. We show that our approach can effectively both (i) fix erroneous answers, and (ii) insert new long-tailed information in LLaVa in a VQA task.

### 5.1 MULTEDIT

Given an image-question $(\mathbf{x}, y)$, we denote a MLLM's generated answer as $O$. MULTEDIT updates a few parameters in the model such that it generates a new answer $O^*$. Similar to [23, 13], we update the $W_{proj}^{\ell}$ matrix at specific causal MLP layers such that the probability $\mathcal{P}(O^*)$ increases. Specifically, we view $W_{proj}^{\ell}$ as a linear-associated memory (where the matrix's input are treated as keys and its output as values) and uses a closed-form update to map the original keys to new (correct) values. Below, we outline the process for acquiring both the keys and values.

**Obtaining keys.** Let $\{v_i\}_{i=1}^{N}$ be the visual token embeddings and $\{e_i\}_{i=1}^{M}$ the text token embeddings. Given a causal layer $\ell$ and the last token of a constraint $c$, we refer to the input of the layer's $W_{proj}^{\ell}$ matrix as its keys. Specifically, we define the key $k_{c,\ell}$ to be the input embedding to the $W_{proj}^{\ell}$ matrix corresponding to the last token of the constraint. This can be obtained with a simple forward pass with the visual and text token embeddings.

**Obtaining values.** Given a causal layer $\ell$, we refer to the output of the layer's $W_{proj}^{\ell}$ matrix as its values. Specifically, we define $z_{c,\ell}$ as the output embedding corresponding to the last token of the constraint. We optimize $z_{c,\ell}$ such that the probability of the correct answer $\mathcal{P}(O^*)$ increases as:

$$z_{c,\ell}^* = \arg\min_{z_{c,\ell}} \mathcal{L}(z_{c,\ell}) \tag{3}$$

where $\mathcal{L}(z_{c,\ell})$ is the standard next-token prediction loss used to train LLMs:

$$\mathcal{L}(z_{c,\ell}) = -\log \mathcal{P}(O^*|v_1...v_N e_1....e_M) \tag{4}$$

MULTEDIT modifies the $W_{proj}^{\ell}$ matrix such that the old keys $k_{c,\ell}$ are mapped to the new optimized values $z_{c,\ell}^*$ which increase $\mathcal{P}(O^*)$. We define MULTEDIT's editing objective as:

$$W_{proj}^{\ell*} = \arg \min_{W_{proj}^{\ell}} \|W_{proj}^{\ell} k_{c,\ell} - z_{c,\ell}^*\|_2^2 + \lambda \|W_{proj}^{\ell} - W_{proj}^{\ell'}\|_2^2 \tag{5}$$

The second regularization term ensures that $W_{proj}^{\ell'}$, the weights before the edit, do not deviate too much from $W_{proj}^{\ell}$. This helps to preserve performance on unrelated VQA pairs.

## 5.2 Experimental details

We experimentally validate MULTEDIT in two real-world model-editing applications:

**Fixing incorrect answers to common questions.** We test MULTEDIT on a set of ∼450 visual questions which LLaVa answers incorrectly (detected using incorrect VQA accuracy) from the multi-modal Known dataset in *VQA-Constraints*. These are generally questions about well-known places, persons and brands and companies.

**Inserting long-tailed VQA knowledge.** We test MULTEDIT on a set of visual questions from the Encyclopedia-VQA dataset [25]. These query fine-grained knowledge about rare landmarks around the world, which MLLMs have been shown to struggle on [25].

For both settings, we compare MULTEDIT to the fine-tuning baselines:(i) fine-tuning from [23] which fine-tunes all the layers using the language modeling objective, and (ii) fine-tuning with constraints from [38] which fine-tunes the layers in a language model with a constraint on the weights to ensure local loss continuity.

We measure the success of each edit operation using the following metrics: (i) **Editing Efficacy**, which uses $\mathcal{P}(O^*)$ to measure the edited model's ability to generate the correct answer for image-question $(\mathbf{x}, y)$. (ii) **Generalization**, which uses $\mathcal{P}(O^*)$ to measure the edited model's ability to generate the correct answer for the question $y$ paraphrased using a language model (see Appendix for details), and (iii) **Specificity**, which uses VQA accuracy [1] to measure the edited model's performance on unrelated VQA questions. We consider unrelated questions to be those from the OK-VQA and Movies datasets in *VQA-Constraints* (see further details in Appendix G).

## 5.3 Results

Overall, our results show that updating the projection matrix using MULTEDIT at just a single early (causal) MLP layer can be a very effective approach for both correcting incorrect answers and inserting new knowledge in MLLMs.

**Fixing incorrect answers.** In Fig. 6 (left), we show that MULTEDIT is able to successfully fix the generations for all questions with wrong answers. In editing efficacy, the average probability of the correct answer improves from 0.07 to 0.82 after the edit. We also observe strong generalization, with a probability of 0.76 for the correct answer even when the question is paraphrased. Although we see a small drop of $1.5\%$ accuracy on unrelated image-question pairs, we note that MULTEDIT outperforms fine-tuning with and without constraints on all the metrics.

**Inserting long-tailed information.** In Fig. 6 (right), we show that MULTEDIT is able to reliably insert new long-tailed knowledge in the model with an editing efficacy of 0.83. We see similar strong generalization when paraphrasing questions with an efficacy of 0.79. Similar to above, MULTEDIT incurs a small drop of $1.4\%$ on unrelated questions but is less affected than other methods.

In Fig. 14, we also provide ablations showing that editing the early causal MLP layers leads to better editing efficacies than editing the middle or the later MLP layers.

## 6 Conclusion

Our paper takes a closer look at how MLLMs process multi-modal information. We contribute a novel multi-modal causing tracing methodology and test-bed, *VQA-Constraints*, as well as a range of novel insights on how MLLMs retrieve and transfer information. We also introduce a novel model-editing algorithm, MULTEDIT, which can effectively fix errors or introduce long-tailed knowledge in MLLMs using a simple closed-form update which targets the early causal MLPs. Overall, this work deepens our scientific understanding of recent MLLM architectures, and enables future work in this direction.

## Author Contributions

Samyadeep Basu conceived the idea, implemented the algorithms, curated the dataset, ran experiments and wrote the paper. Daniela Massiceti provided critical feedback at every stage of the project (algorithms, dataset) and worked with Samyadeep to write the paper. Besmira Nushi, Soheil Feizi and Cecily Morrison provided critical feedback on the project direction, paper writing and presentation. Martin Grayson helped the team with experiments on curating the probe dataset.

## Acknowledgements

This project was supported in part through a part-time internship at Microsoft Research and in part was supported in part by a grant from an NSF CAREER AWARD 1942230, ONR YIP award N00014-22-1-2271, ARO's Early Career Program Award 310902-00001, Army Grant No. W911NF2120076, the NSF award CCF2212458, NSF Award No. 2229885 (NSF Institute for Trustworthy AI in Law and Society, TRAILS), an Amazon Research Award and an award from Capital One. We thank Hamid Palangi for proofreading the draft and providing feedback.

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

## A  Limitations and Ethical Considerations

The findings of our work are limited to factual questions on natural images with short-form answers. Future work should investigate other types of questions (e.g. subjective), image domains (e.g. charts), and longer-form answers. Our work also does not map visual tokens to specific concepts in the input image – another important research direction. Our editing method, MULTEDIT, while effective for correcting and adding new information, could also be used to add false or harmful information into a MLLM. Work is needed on how to reliably detect misinformation in multi-modal settings.

## B  *VQA-Constraints* Details

In Sec. 3.1, we introduce the constraint framework for the task of factual VQA. In particular, there are two types of constraints: (i) Visual Constraint: A set of words in a question which refer to an entity in an image; (ii) Text Constraint: A set of words in a question which together with the visual constraint are responsible for answering a question. In our *VQA-Constraints* dataset there are two types of questions: (i) Single Constraint Questions: These questions comprise of only the visual constraint; (ii) Multi-Constraint Questions: These questions comprise of both the visual as well as the text constraint. In our paper, we primarily focus on the Single Constraint questions, therefore a majority of questions in our test bed of *VQA-Constraints* consist of only the visual constraints. Our VQA dataset *VQA-Constraints* which is annotated with constraints comprise of the following three parts:

**OK-VQA**: We annotate the questions in the test-set of OK-VQA with the constraints. In total, this part consists of 5k questions. We use the same images as those used in the original OK-VQA test-set.

**Multimodal Movies**: We use the text-only WikiMovies dataset from [37] and download images using the Bing API using the name of the directors. Our team filters and validates that the downloaded images are of the director itself and there's no noise in the downloaded images. We use a set of 1.5k questions in the final dataset.

**Multimodal Known**: For understanding knowledge storage in language models, [23] use a probe dataset (known.json) consisting of 1.2k questions. We first replace the subject in the question with a constraint and then download images from Bing API. Our team validates that the downloaded images are correct and uses it to create the Multimodal Known dataset.

**Multi-Constraint Questions**: The multi-constraint questions in *VQA-Constraints* are created from OK-VQA and Multimodal Movies. In particular, there are 150 multi-constraint questions for OK-VQA and 500 multi-constraint questions from Multimodal Movies. Given that the primary focus of our paper is on Single Constraint questions, this set of Multi-Constraint questions is relatively small.

In total, *VQA-Constraints* consist of 9.7K VQA questions with Single Constraints and 650 Multi-Constraint VQA questions which we use for interpretability analysis.

| Dataset | Description | Example 1 | Example 2 | Example 3 |
|---|---|---|---|---|
| | | | Description of Questions in VQA-Constraints | |
| OK-VQA | Single Constraint | What sport can you use *this* for? | Why might someone go to *this place*? | What flavor is *this pastry*? |
| OK-VQA | Single Constraint | Which airlines is this insignia? | What fruits come from *these trees*? | What is *this animal* mostly known for? |
| Multimodal Known | Single Constraint | *This person* is a citizen of? | *This newspaper* is written in? | The capital of *this city* is in? |
| Multimodal Known | Single Constraint | *This venue* is owned by? | *This building* is located in which city? | *This show* debuted on? |
| Multimodal Movies | Single Constraint | Name a movie directed by *this director*? | Name a movie directed by *this director*? | Name a movie directed by *this director*? |
| Multimodal Movies | Single Constraint | Name a movie directed by *this director*? | Name a movie directed by *this director*? | Name a movie directed by *this director*? |
| OK-VQA | Multi-Constraint | Which is the most famous type of *this food* in *India* | How does the population of *this city* compare to *Mumbai*?" | What is the relationship between *this person* and *Venus Williams*? |
| Multimodal Movies | Multi-Constraint | Name a movie directed by *this director* in year *2006*? | Name a movie directed by *this director* in year *2010*? | Name a movie directed by *this director* in year *1994*? |

Table 1: **Description of the different VQA questions in** *VQA-Constraints*. The constraints are marked in *italic* in the columns.

**Annotating and Validating the Visual Constraints.** To automatically annotate the constraints in the visual questions from *VQA-Constraints*, we use a strong language model such as GPT-4. In particular, we annotate only the Multimodal Known and OK-VQA sub-parts of the *VQA-Constraints* dataset. First from each dataset, we annotate 100 examples which we use as in-context examples to the language model to annotate new questions. Given these annotations, our team then verified if the annotation is correct and if incorrect, modified the constraint annotation to make it correct. In total, the visual constraints are annotated for 9.7k VQA questions.

## C  Standard Causal Tracing Does Not Recover Causal States

In Fig. 7, we find that the standard causal tracing approach, which adds Gaussian noise to the span of visual tokens and the constraint in the text does not recover any causal states, even with a large window size of 10. This is true for recovering both the MLP as well as the self-attention causal layers.

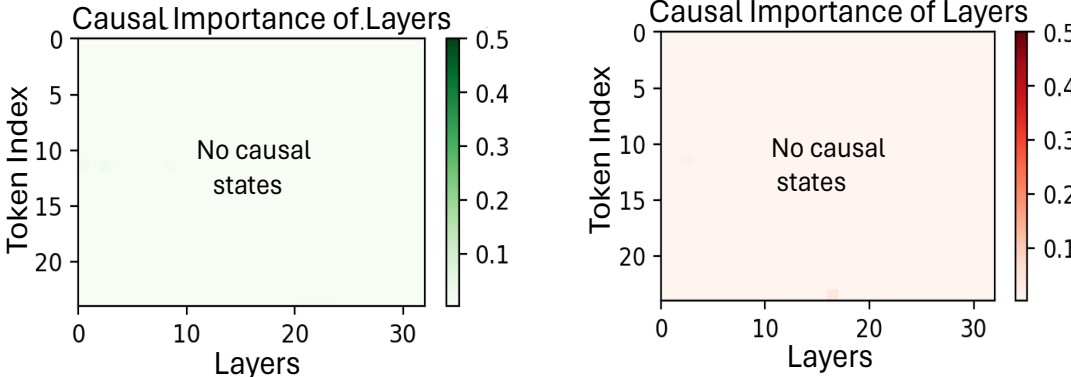

Figure 7: **Standard Causal Tracing**: We find that using the standard causal tracing procedure from [23] is not able to recover causal states. Averaged across all the datapoints in Multimodal Known and Multimodal Movies from *VQA-Constraints*. A window size of 10 is used.

## D  MULTIMODALCAUSALTRACE - Qualitative Plots

In this section, we present various qualitative plots corresponding to MULTIMODALCAUSALTRACE showing that our method is able to recover causal states in both the MLP as well as self-attention layers very early on in the model. We note that even using a small window size of 1 is sufficient to recover causal states. We highlight that earlier works [23, 24] show that for language models, a larger window size of 10 is required and also the middle layers are used to recover the relevant knowledge. However, we find that in the presence of visual prompts, relevant knowledge is retrieved from the early layers.

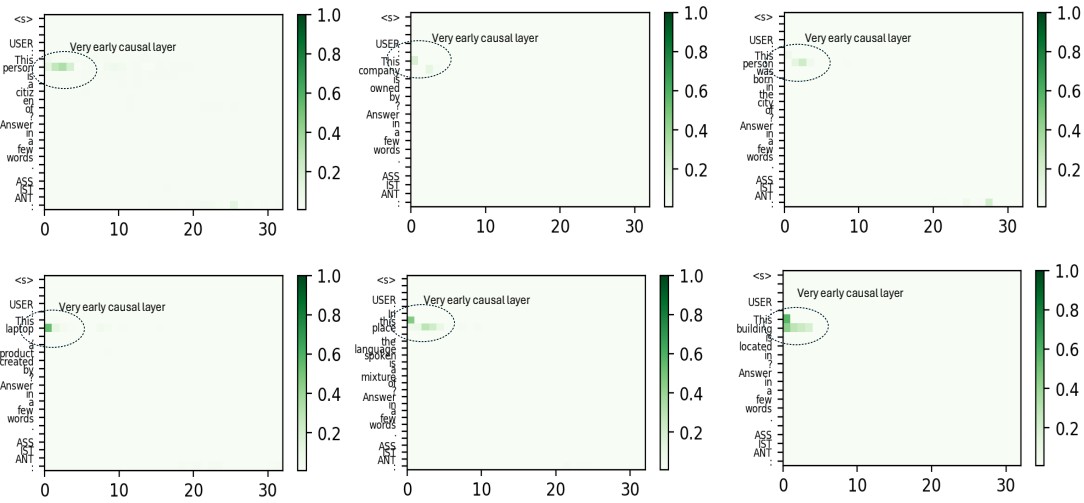

Figure 8: **MULTIMODALCAUSALTRACE with Window Size 1: Early MLP layers are causal.** Qualitative tracing results for examples with single constraint questions (containing only a visual constraint).

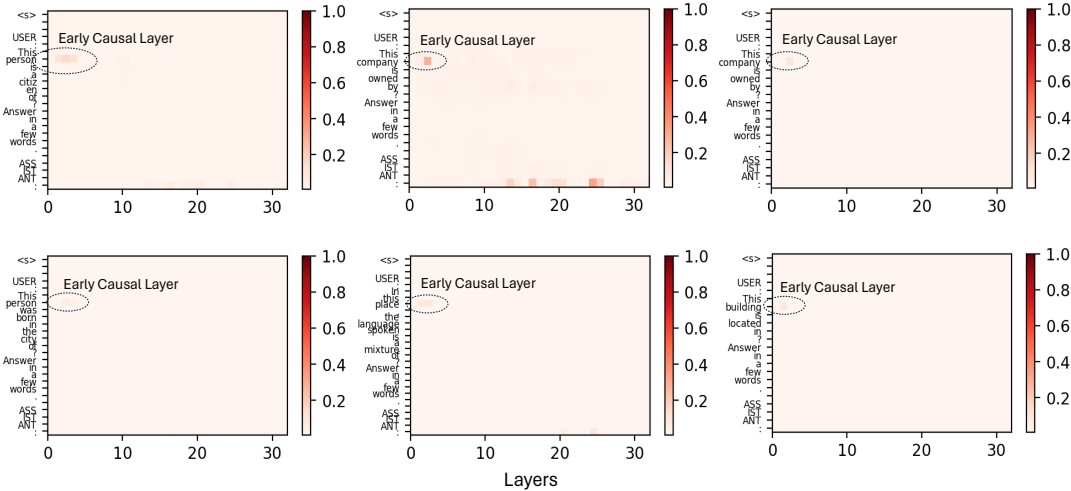

Figure 9: **MULTIMODALCAUSALTRACE with Window Size 1: Early self-attention layers are** *weakly* **causal compared to causal MLPs.** Qualitative tracing results for examples with single constraint questions (containing only a visual constraint).

## E    Multimodal Phi-2 Tracing Results

We use MULTIMODALCAUSALTRACE on multimodal phi-2, where the language model is phi-2 [17], a small language model capable of obtaining strong performances and often close to large language models for certain language tasks. Similar to our earlier observations for LLaVa, we find that even in multimodal phi-2, the early MLP and self-attention layers are causal. This shows the generalizability of our results for latest multimodal language models. We also note that a small window size of 1, is sufficient to recover causal states in multimodal language models.

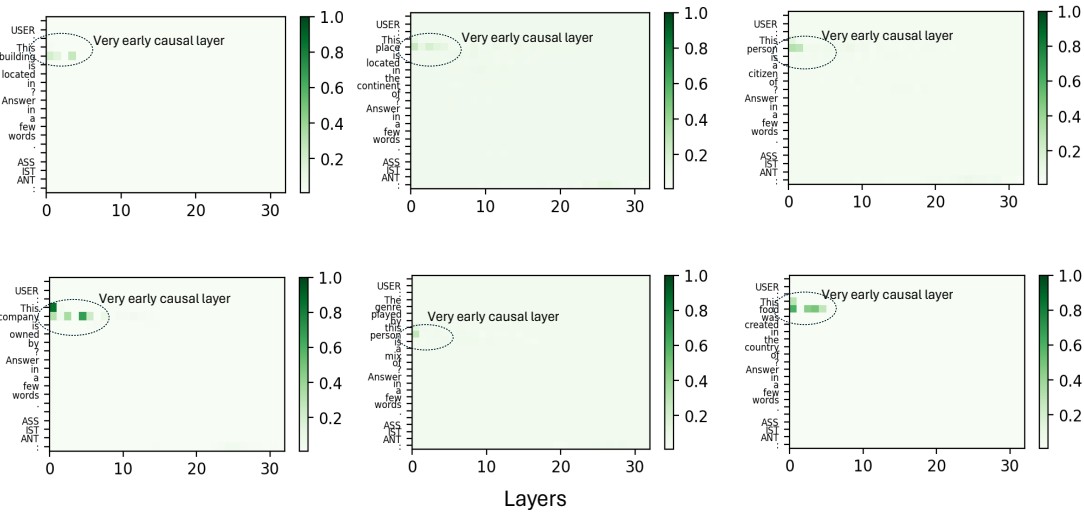

Figure 10: **MULTIMODALCAUSALTRACE with Window Size 1: Early MLP layers are causal for Multimodal Phi-2, similar to LLaVa-7B.** Qualitative tracing results for examples with single constraint questions (containing only a visual constraint).

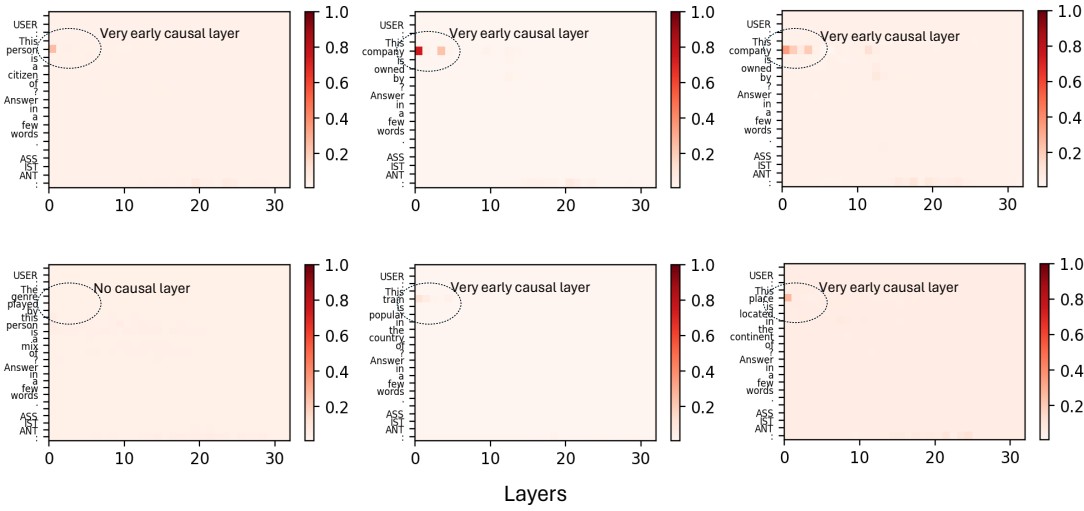

Figure 11: **MULTIMODALCAUSALTRACE with Window Size 1: Early Self-Attention layers are also causal for Multimodal Phi-2, similar to LLaVa-7B.** Qualitative tracing results for examples with single constraint questions (containing only a visual constraint).

## F  Multi-constraint Questions Plots

In this section, we present qualitative plots for the multi-constraint questions from OK-VQA. These questions consist of a visual constraint as well as a text-constraint which in conjunction are used to answer a given question. Given that we earlier observe early causal layers corresponding to the visual constraint – in this section, we show results corresponding to the text-constraint. We have two primary observations: (i) A larger window size (at least a size of 6) is required to recover causal states, which highlights that a set of internal layers are used to retrieve relevant information; (ii) The causal layers span the early as well as the middle layers. This is true for both the MLP as well as self-attention layers.

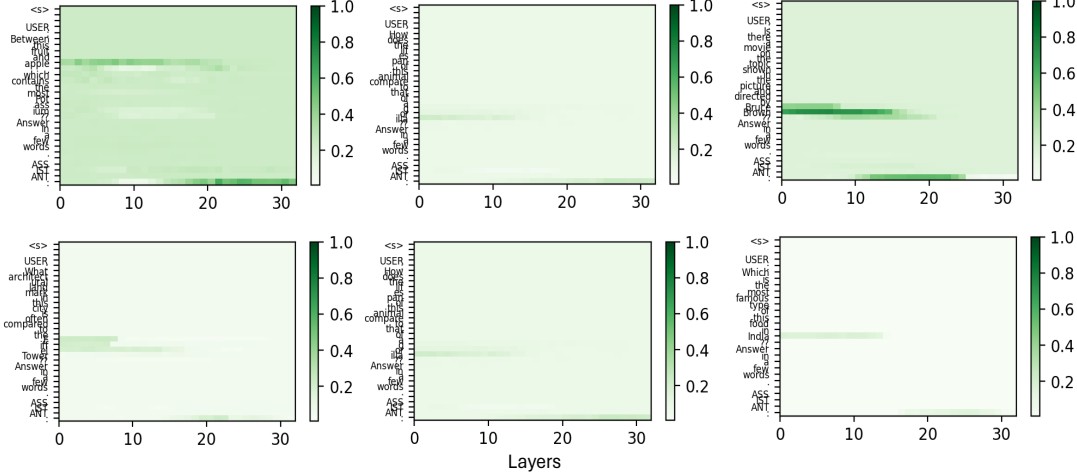

Figure 12: **MULTIMODALCAUSALTRACE with Window Size 6: Early and mid MLP layers are causal.** Qualitative tracing results for examples with multiple constraint questions (containing a visual constraint and a text constraint). Here MULTIMODALCAUSALTRACE is performed at the text-constraint position. For multi-constraint questions, a larger window size is needed to recover any causal states.

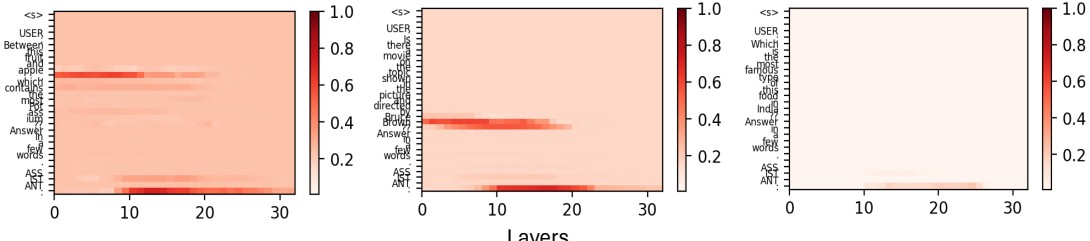

Figure 13: **MULTIMODALCAUSALTRACE with Window Size 6: Early and mid self-attention layers are causal for a majority of the visual questions.** Qualitative tracing results for examples with multiple constraint questions (containing a visual constraint and a text constraint). Here MULTIMODALCAUSALTRACE is performed at the text-constraint position. For multi-constraint questions, a larger window size is needed to recover any causal states.

# G   Model Editing Formulation and Further Results

As described in Sec. 5.1, MULTEDIT modifies the $W_{proj}^{\ell}$ matrix such that the old keys $k_{c,\ell}$ are mapped to the new optimized values $z_{c,\ell}^*$ such that the new values can drive the model towards increasing $\mathcal{P}(O^*)$. We define the editing objective in MULTEDIT as below:

$$W_{proj}^{\ell*} = \arg \min_{W_{proj}^{\ell}} \underbrace{\|W_{proj}^{\ell}k_{c,\ell} - z_{c,\ell}^*\|_2^2}_{term1} + \underbrace{\lambda \|W_{proj}^{\ell} - W_{proj}^{\ell'}\|_2^2}_{term2} \tag{6}$$

where $W_{proj}^{\ell'}$ denotes the weights of the $W_{proj}^{\ell}$ matrix before the edit optimization. The second term acts as a regularization which ensures that $W_{proj}^{\ell'}$ does not deviate too much from $W_{proj}^{\ell}$.

From Eq. (6), we can observe that it can be solved using a simple closed form update as follows:

$$W_{proj}^{\ell*} = (\lambda W_{proj}^{\ell'} + z_{c,\ell}^* k_{c,\ell}^T)(\lambda I + k_{c,\ell} k_{c,\ell}^T)^{-1} \tag{7}$$

Given access to the keys and values, this closed-form optimization can be solved on a CPU. We note that this closed-form update for model editing has been recently used for text-to-image models [4, 5]. However to the best of our knowledge, ours is the first work to use such an update for MLLMs.

**Hyperparameters.** We use a learning rate of 0.1 to optimize for the values using Adam Optimizer. For the regularization factor $\lambda$, we use 0.01 after a grid search. Amongst the set of early causal layers between 0-5, we find editing Layer 2 to result in the best editing efficacy.

In Fig. 14, we show that editing the early causal MLP layers

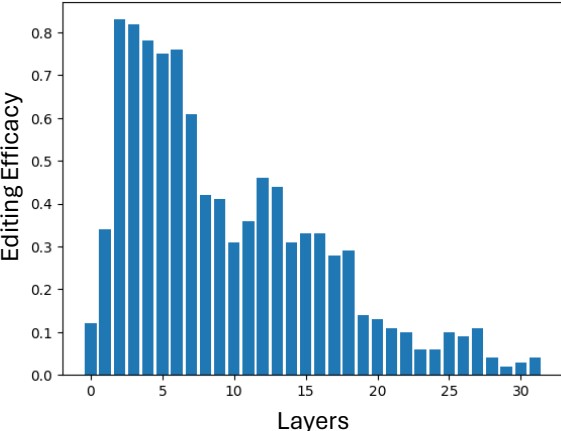

Figure 14: **Editing early causal layers leads to better editing efficacy, though editing non-causal layers also leads to non-trivial editing efficacy. Similar results for language models have been highlighted in [10]**. The editing is performed at the position of the last token of the visual constraint.

is better than the middle or the later layers. However, we find that editing the middle or the later layers also lead to a non-trivial editing accuracy, though it's lesser in terms of effectiveness than editing the early causal layers. Recent work [10] also observe this behaviour of non-trivial editing accuracy at locations which are not causal layers.

# H   Information Transfer and Early Failure Mode Detection - Qualitative Plots

In this section, we first show qualitative in Fig. 15 that the attention contributions from the constraint tokens to the last token is significantly more for correct answers than for incorrect answers. This is quantitatively highlighted and shown in Fig. 16 where we find that Layer 16 and Layer 17 to be the main orchestrator of this difference. A recent work [37] use attention contributions as a signal to detect hallucinations in language models. They show that the attention contribution values when used together with a linear model can be used to predict the correctness of the answer. In fact, they find this metric to be close to the confidence metric (which is the probability of the generated token). However, [37] necessitates the use of all self-attention layers which reduces it to be used as an "early" failure mode detection metric. For multimodal language models, instead of using all the self-attention layers, we use the average attention contributions from Layer 16 and Layer 17 as an "early" failure mode detection metric. From Fig. 17, we find that the average attention contributions indeed lead to a non-trivial AUROC of 0.63, but it lags behind the confidence metric which obtains AUROC of 0.76. Our early results show that using the model internals can be used to "early" detect if the model is going towards a failure mode, but it lags behind the confidence although that requires the full forward pass.

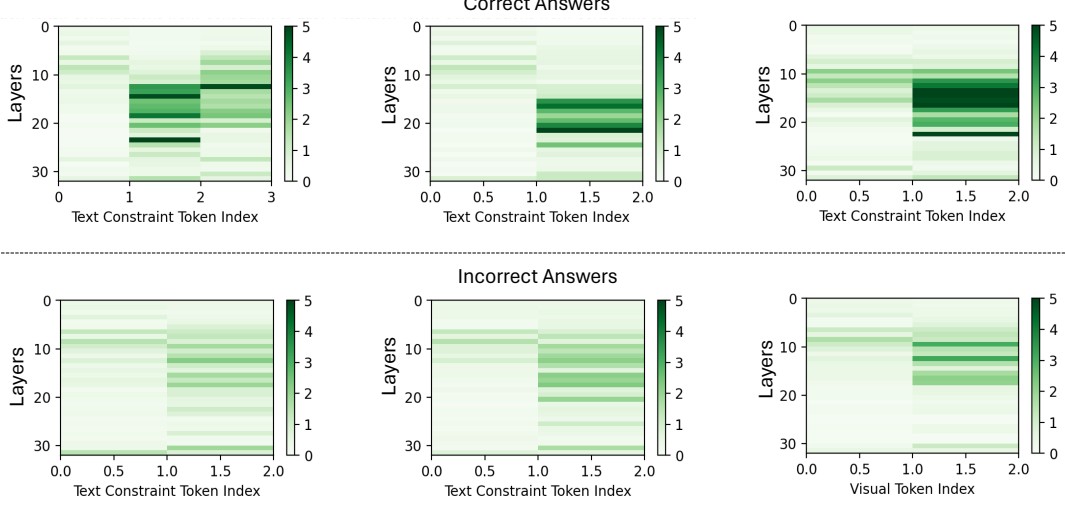

Figure 15: **Correct answers have stronger attention contributions from the constraint tokens (X-axis) to the last token, when compared to incorrect answers**. These qualitative examples are from the Multimodal Known dataset in *VQA-Constraints*.

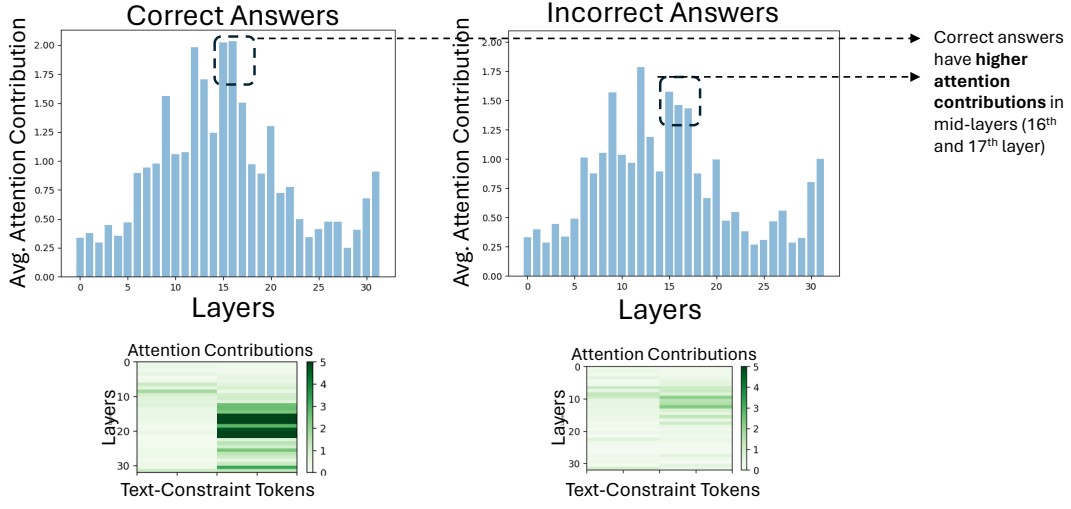

Figure 16: **Correct answers have stronger attention contributions from the constraint tokens (X-axis) to the last token, when compared to incorrect answers on an average**. Computed on the Multimodal Known dataset in *VQA-Constraints*. In particular the layer 16 and layer 17 are the distinguishing layers which have higher attention contributions in the correct answers.

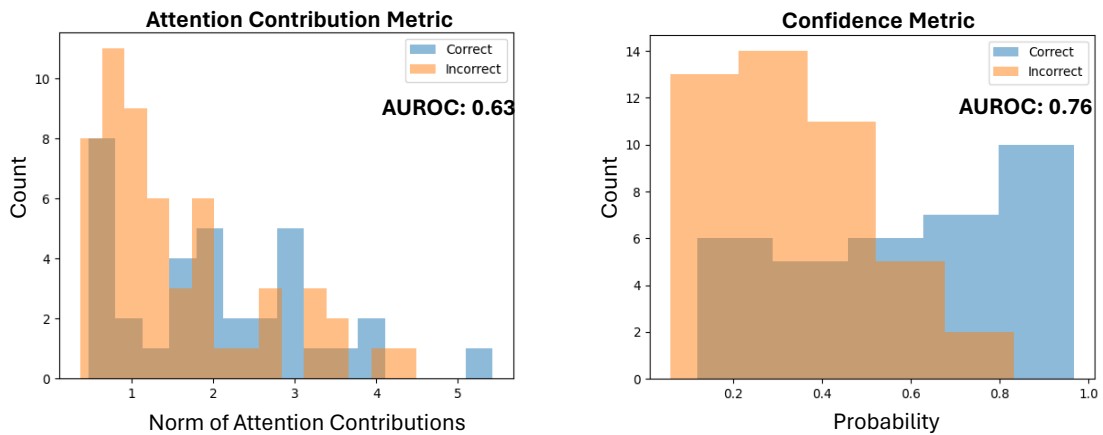

Figure 17: **Attention Contribution Metric although lags behind Confidence Metric, it can be used as a coarse "early" failure mode detection metric**. Dataset used: Multimodal Known from *VQA-Constraints*. We use the average attention contributions from Layer 16 and Layer 17 for computing the failure mode metric.

# I   Editing Dataset

The editing dataset consists of two parts: (i) VQA questions from Multimodal Known on which the LLaVa-7B gives incorrect answers; and (ii) VQA questions from Encyclopedia-VQA which tests the long-tailed knowledge of the model. The long-tailed questions are primarily about landmarks. Below we present some qualitative examples on the long-tailed VQA questions:

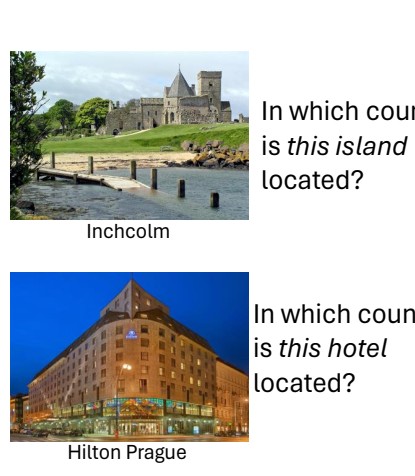 In which country is *this island* located?

Inchcolm

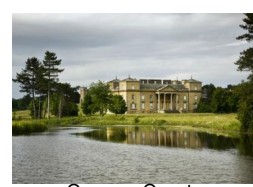 In which country is *this house* located?

Croome Court

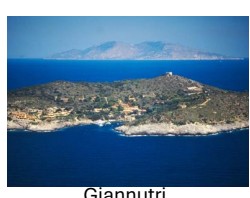 In which country is *this hotel* located?

Hilton Prague

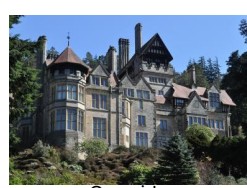 In which country is *this house* located?

Cragside

In which country is *this island* located?

Giannutri

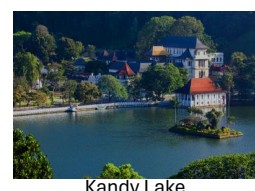 In which country is *this lake* located?

Kandy Lake

Figure 18: **Qualitative Examples from the long-tailed knowledge editing subset.** The questions are sourced from the questions involving landmarks in Encyclopedia-VQA.

## J   Compute

All our experiments are performed on Nvidia-A6000 and A5000 GPUs.

