# OpenReview forum: "Understanding Information Storage and Transfer in Multi-Modal Large Language Models"
_NeurIPS.cc/2024/Conference — NeurIPS 2024 poster_

### Official Review · Reviewer_eXiq · 2024-06-24

**Soundness:** 3
**Presentation:** 3
**Contribution:** 2
**Rating:** 5
**Confidence:** 3

**Summary:**

This paper presents an approach to investigate the layer in which multi-modal large language models retrieve factual information and show several insights into their behavior. To investigate causal tracing, they propose replacing the input text tokens with different ones so that the model can respond differently from the correct answer with the replaced text tokens, then they copy activations from the clean layers until the model can reconstruct the correct answer. Through this investigation, their primary finding is that information to answer a visual question is mainly retrieved from early MLP and self-attention layers in MLLMs, which is different from the insight that LLMs retrieve factual information from middle layers. They also reveal several facts: only a subset of visual tokens are involved in sending information to the LLM's early layers, and mid-layer self-attention layers are involved in transferring information from the early causal layers to the question’s final token. They also demonstrate that an approach similar to editing factual knowledge of LLMs can be used to overwrite the knowledge of MLLMs.

**Strengths:**

1. Their main strength is providing an approach to locate the layers in which MLLM models are retrieving factual information. Since the research on MLLM is popular these days, the technique may be a good one to analyze the behavior of MLLM.

2. They present a new dataset to study the issue.

3. The insight obtained from their approach is interesting. Unlike LLMs which retrieve factual information from middle layers, MLLMs retrieve factual information from early layers. This fact can be new to the community.

4. Other observations such as how visual tokens are used in MLLMs are insightful too. These facts might be useful to think about the design of MLLMs.

**Weaknesses:**

1. They offer an interesting observation that MLLM retrieves factual information from early layers, but they lack insight into why this happens. I think it is very important to give intuitions on why their behavior is different from LLMs.

2. They present the insight from their approach using specific examples as in Fig. 3. However, if my understanding is correct, they do not provide numerical stats of which layers are responsible for retrieving factual information. Since they construct a new dataset, I guess it is easy to compute quantitative values to compare in which layers LLM and MLLM are retrieving information.

3. The overall idea of their approach is not very novel. They borrow the idea from prior work and adapt it to MLLM. The novel part of their approach is replacing input text tokens so that the model makes an incorrect answer.

4. I think section 5 does not improve the value of this submission. The section focuses on how to edit the knowledge of MLLM following prior work. The topic is related, yet different from their core contributions.

**Questions:**

At this phase, I am on the borderline. I acknowledge the importance of analyzing the behavior of MLLMs and think the proposed approach is simple and reasonable. But, I also think the findings from this paper are not very novel.

1. Related to 2 in weaknesses, is the value of Fig. 3 all from a specific example? or is it averaged over many examples?

2. Please respond to all the weaknesses above since I might misunderstand some parts.

**Limitations:**

Yes.

---

> ### Author Rebuttal · Authors · 2024-08-06
>
> We thank the reviewer for their constructive feedback. We are glad that they acknowledge the strength and importance of our proposed methodologies and findings for future MLLM development and interpretability. Below we address the specific points raised by the reviewer:
>
> **They offer an interesting observation that MLLM retrieves factual information from early layers, but they lack insight into why this happens. I think….**: Through conducting our analyses, we developed several hypotheses as to why MLLMs retrieve information differently compared to LLMs. We suspect that the language model elicits distinct internal circuits (a set of MLPs and attention heads) in the presence and absence of visual information. If the visual information is not present, the language model relies on one type of circuit, but in the presence of information from visual tokens, this circuit gets overridden by another. This could be because the visual tokens contain fine-grained information about the constraint that forces the model to take a different path to obtain the final answer. Given the circuits are different, the nodes (e.g., MLPs/attention heads) in the circuit will therefore also be different, hence we see different causal layers activated for MLLMs versus LLMs.
>
> Validating these hypotheses fell outside the scope of this paper, however, our work lays down a strong foundation towards such future work. We will add a discussion on these hypotheses in the Appendix of the final version of the paper and lay down some possible directions for future work.
>
> **They present the insight from their approach using specific examples as in Fig. 3. However, if my understanding is correct, they do not provide numerical stats of which layers are...is the value of Fig. 3 all from a specific example?**: Fig. 3 is averaged across the examples in VQA-Constraints, thus providing a robust (visual) estimate of the causal layers in MLLMs for a factual VQA task. As suggested by the reviewer, we will augment this plot with numerical statistics to improve the readability of the results in the final version of the paper.
>
> **The overall idea of their approach is not very novel. They borrow the idea from prior work and adapt it to MLLM.**: Our proposed methodology, MultiModalCausalTrace, proposes a non-trivial adaptation to existing causal tracing methodologies in order for it to work for multi-modal inputs. Without it, no causal traces would have emerged (see Fig 7 - Appendix where we show this result) and our subsequent insights on MLLM information storage and retrieval would not have been possible.
>
> We also note that building on existing methodologies, even in small ways, drives the field meaningfully forwards. For example, the Vision Transformer is different from the Transformer only in how it views image inputs as a grid of visual tokens, yet this architecture has unlocked enormous advances across a wide range of computer vision tasks. We believe that MultiModalCausalTrace along with our probe-dataset can similarly unlock rich mechanistic interpretability studies for MLLMs in the future.
>
> **I think section 5 does not improve the value of this submission. The section focuses on how to edit the knowledge of MLLM following prior work. The topic is related, yet different from their core contributions**: Thank you for the feedback! We included Section 5, due to the following reasons:
> 1) First, it allowed us to validate the multi-modal causal tracing methodology proposed in Section 4. If our identified causal traces were spurious or not meaningful, then any subsequent model editing would have immediately revealed this.
> 2) Second, it introduces (and validates) one practical application where these causal traces could be used in the real-world. Correcting errors and adding new information has been widely explored for LLMs in light of the huge costs (including environmental) that are associated with re-training or fine-tuning these models. We, therefore, felt it was an important direction to direct the research community’s attention towards. Interestingly, our experiments suggest that targeted MLLM model editing is a better alternative to fine-tuning which we hope will inform future research in this direction. The finding is particularly relevant for long tail knowledge, for which it may not even be possible to have sufficient data for training or fine tuning.

---

### Official Review · Reviewer_BKyn · 2024-07-12

**Soundness:** 3
**Presentation:** 3
**Contribution:** 2
**Rating:** 6
**Confidence:** 3

**Summary:**

The paper introduces MULTIMODALCAUSALTRACE, an algorithm designed to identify hidden states in large language models (LLMs) that store factual information, specifically extending this capability to vision-language models where images are encoded as visual tokens. Building on previous work in information storage identification, MULTIMODALCAUSALTRACE constructs three models: a clean model, a corrupted model (where visual tokens are randomly replaced, perturbing the hidden states), and a restored model (where some corrupted hidden states are replaced with clean ones). By conducting mediation analysis on the causal relationship between state corruption and next-token prediction outcomes, the algorithm can pinpoint layers associated with different facts. The study reveals that multi-modal LLMs exhibit distinct behaviors in information storage and propagation, such as storing facts in middle layers rather than earlier transformer layers.

Additionally, the paper contributes the VQA-constraints dataset, derived from existing visual QA datasets and annotated with constraints.

The paper also proposes MULTEDIT, an algorithm for inserting or correcting factual knowledge in multi-modal LLMs. MULTEDIT optimizes the projection matrix in the MLP layers of transformers to minimize the mean squared difference between the projection output and the value vector that maximizes the prediction probability of the desired output. Empirical results demonstrate that MULTEDIT effectively corrects factual errors in VQA tasks.

**Strengths:**

+ The paper tackles an important task of identifying and correcting factual information in visual language model, a topic that received relatively less attention than in language modeling.

+ The paper provided strong motivation to the problem and background it addresses, as well as clear visual and formal presentation of the methodology adapted.

+ The proposed solution is comprehensive and encompassed various aspect of the information storage analysis problem. It delivers valuable insights to some practical challenges of adapting existing LLM-based methods to vision-language models such as the increased number of constraint tokens due to visual encoding.

**Weaknesses:**

+ The uniqueness and impact of multi-modal models are not thoroughly demonstrated. As stated in the paper, the proposed methods is largely based on the canonical Causal Trace(CT) and Rank-One Model Editing (ROME), with modifications in implementation allowing them to work for vision-language models. However, the choice (or absence) of most of such modifications are not well-motivated and no ablation study provided to understand their exact contribution. For example, the proposed MULTEDIT largely resembles the existing ROME method, with differences in how the key vector is found and how the optimal value vector is acquired. But it is not clear from the text why such changes are necessary and/or make the method works better for visual language model.

+ The lack of discussion on multi-modal properties hinders the paper's novelty. The overall structure of the paper is similar to [23], and the main text does not substantially differentiate the methodology from prior work.

+ Minor formatting issue in the reference: journal/conference names are missing in some items (such as [23] [24])

(The reference indices of the original submission are used throughout this review)

**Questions:**

+ In vanilla ROME editing, the value vector is found by optimizing both the probability of the desired output and similarity to the essence of the original sentence. But in MULTEDIT the later term is dropped. What is the purpose of such change and how does the updated objective preserve the understanding of the overall prompt?

+ In the creation of VQA-Constraints, does the authors manually correct all annotations generated by GPT-4? If so, what is the main advantage of starting with automated annotations?

**Limitations:**

The authors included a limitation section in the appendix covering drawbacks of the proposed method and potential negative societal impacts.

---

> ### Author Rebuttal · Authors · 2024-08-06
>
> We thank the reviewer for their constructive feedback, and for acknowledging the comprehensiveness of our findings and insights. We are glad that they also see the importance of being able to identify and correct factual information in MLLMs, which is currently an understudied problem. Below we address the points raised by the reviewer:
>
> **“The uniqueness and impact of multi-modal models are not thoroughly demonstrated…..’**: Our two methodologies, i) MultimodalCausalTrace for multi-modal causal tracing, and ii) MultEdit for multimodal editing method, are novel in their ability to work with multimodal (image-text) inputs. Towards i), we found that applying existing causal tracing methodologies out-of-the-box did not elicit relevant causal traces for multimodal inputs (attributed to large noise in the corrupted model due to large number of visual tokens) - see Fig.(7) - Appendix. This required us to extend these methodologies in a non-trivial way for multi-modal inputs. Towards ii) we introduce a novel editing approach that is based on ROME, but improves it with an updated loss objective that does not rely on caching a Wikipedia dataset. This is computationally preferred in a multi-modal setting, as images have a larger memory footprint than text alone. We also introduce a novel probe dataset, VQA-Constraints, to conduct our analyses and can be used to drive future impact in multimodal interpretability studies.
>
> **As stated in the paper, the proposed methods is largely based on the canonical Causal Trace(CT) and Rank-One Model Editing (ROME),.....**: We note that MultEdit has certain distinctions from the ROME method which we detail as follows:  While the high-level idea of employing a closed-form approximation to update (edit) a targeted localized weight matrix in the network is similar to LLMs, there are certain major distinctions in the editing method itself: (i) If looked at closely, the edit-optimization equations in ROME [23] - Eq. (2) is different from Eq. (5) of MultEdit in our paper. In principle, our editing method does not require caching a sample of Wikipedia text for computing the uncentered covariance in their equation. This term in the LLM editing works (e.g., ROME)  ensure that the values corresponding to keys for unrelated texts remain similar. However, in our case, we enforce this condition with the L2 regularizer ensuring that the updated weight matrices do not deviate significantly from the original weight matrices (controlled by a hyperparameter \lambda). We find that this simple modification leads to strong editing results. One can also use the editing equation from ROME to update MLLMs, but that would require caching the embeddings from a multimodal Wikipedia type dataset (which is not readily available and requires curation/cleaning) which might be inefficient and also incur an additional operation.  (ii) We also find that obtaining the values by *only* optimizing the multimodal language modeling next-token prediction loss is sufficient towards obtaining good embeddings of values – which lead to good editing performance on using it with Eq. (5) in our paper.  During our experiments, we did add a KL divergence loss where the objective function was to maintain the output probability distribution with the prompt (visual prompt + <visual-constraint> is a ) between the original MLLM and the MLLM whose value vector is optimized. However, empirically we did not see an improvement in the editing performance.
>
> Overall, MultEdit is simpler in implementation and  does not require caching a multimodal Wikipedia entry while leading to strong editing performances.  We will add these distinctions from the LLM editing works such as ROME in the final version of our paper. We also note that our paper has a package of contributions (including MultimodalCausalTrace and VQA-Constraints) that as a whole advance the understanding of large multimodal language models.  Each of these steps required technical novelty in terms of handling multimodal information and architectures, besides adapting current techniques.
>
> **Minor formatting issue in the reference: journal/conference names are missing in some items (such as [23] [24]):**: Thank you for pointing this out. We will fix these formatting issues in the final version of the paper.
>
> **The lack of discussion on multi-modal properties hinders the paper's novelty. The overall structure of the paper is similar to [23], ...**: We point the reviewer to Section 3.2 in our paper, where we describe the causal tracing methods used for language models and how our method MultiModalCausalTrace differs and extends this to work for multi-modal inputs. In the final version of the paper, we will make the distinctions between Causal Trace and MultimodalCausalTrace more clear for better readability. We also point the reviewer to the first point and second point in the rebuttal where we discuss the distinctions and the uniqueness of the various components in our paper.
>
> **In the creation of VQA-Constraints, does the authors manually correct all annotations generated by GPT-4? If so, what is the main advantage of starting with automated annotations**: For the Movies dataset in VQA-Constraints, the questions are templated (e.g., questions are of the form ‘Name of movie directed by this director’) so the visual constraint (e.g., this director) is defined by default. For the OK-VQA and Known datasets in VQA-Constraints, we conducted an MTurk study to filter and correct erroneous annotations. Initially, we used automated annotations, finding that GPT-4 achieved approximately 96% accuracy in annotating constraints in VQA questions with as few as 50 in-context examples (which we manually annotate with constraints). Based on this high efficacy, we used GPT-4 for the initial annotations and then used MTurk to correct any remaining errors (<3% of the total examples in VQA-Constraints). We will include these filtering details in the final version of the paper.

---

### Official Review · Reviewer_3JTJ · 2024-07-13

**Soundness:** 3
**Presentation:** 3
**Contribution:** 3
**Rating:** 7
**Confidence:** 4

**Summary:**

This paper studies the information storage and transfer for the multi-modality large language model (MLLM). The author provides a comprehensive empirical study leveraging the causal tracking method, i.e., corrupting a clean model by perturbing the input prompt, to identify which layers are used to retrieve information relevant to the constraints in the prompt. The author also leverages the attention contributions to compute how much one set of input tokens influences a set of output tokens to track how information is transferred from visual tokens to the causal layers. To provide new insights, a new dataset called VQA-Constraints has been created to support the empirical study. The author provides many new insights that are different from the existing research in LLM that the MLLMs reply on MLP and self-attention block in much earlier layers for information storage and a consistent small subset of visual tokens output by the vision encoder are responsible for transferring information from the image to these causal blocks. The mechanism revealed in this study also inspires a model-editing algorithm that can correct errors and insert new long-tailed information into MLLMs.

**Strengths:**

1. Overall, this paper is very well written, and the main messages are conveyed smoothly. The findings and takeaway message have been delivered clearly.

2. The paper's novelty stems not only from the research problem but also from the design of the exploration method, the novel insights, and the corresponding proposed model-editing method. To the best of my knowledge, this should be the first work that provides a comprehensive study of knowledge tracing on MLLM, providing a great foundation for future exploration.

3. The research findings are very interesting and insightful. They are validated by different datasets and the newly proposed dataset, making the insights solid and sound.

4. The proposed model-editing method is effective and can partially validate the mechanism identified in the paper.

**Weaknesses:**

This is a strong paper in general, and the review does not realize the critical weakness of the present paper. The reviewer may be curious whether the code will be public upon the acceptance.

**Questions:**

Please refer to the Weaknesses sections for more details.

**Limitations:**

N/A.

---

> ### Author Rebuttal · Authors · 2024-08-06
>
> We thank the reviewer for their positive feedback, in particular acknowledging that this is the “first work that provides a comprehensive study of knowledge tracing on MLLM”. We are also glad that the reviewer finds our paper well-written and feels that the work can provide a solid foundation for future exploration in the space of MLLM interpretability. We will certainly be releasing the code publicly, and will link it in the final camera-ready version of the paper.

---

### Official Review · Reviewer_ieTM · 2024-07-23

**Soundness:** 3
**Presentation:** 4
**Contribution:** 3
**Rating:** 7
**Confidence:** 4

**Summary:**

This manuscripts studies mechanistic interpretability in autoregressive vision-language models. Towards this, the authors propose `MultiModalCausalTrace`, an extension of the causal tracing technique for analyzing text-only LLMs, which is to perturb "visual constraint" tokens with a set of semantically coherent, but irrelevant tokens, and measure its change effects in model behavior. The authors observe that VLMs store and transfer information at early layers, while LLMs operate in early-to-mid layers. Based on this observation, the paper proposes `MultEdit`, a technique that injects long-tailed information in these causal layers of VLMs.

**Strengths:**

- The paper's presentation is very clear and tells a coherent story.
- The main techniques (multimodal causal tracing) are reasonable; they are built upon well-tested frameworks for mechanistic interpretability of LLMs.
- The findings are intriguing as the authors observe VLMs behave differently in terms of information storage and transferring, compared to LLMs.
- `VQA_Constraint` is a valuable contribution to the interpretability community, and binding visual input to a natural language reference is a reasonable way to evaluate the knowledge for VQA, and is different from the many algorithmic tasks studied in prior mechanistic interpretability works on text-only LLMs.

**Weaknesses:**

- While `MultEdit` has demonstrated impressive editing efficiency and generalization performance, the technical novelty is somewhat limited  as it is an application of a well-tested technique for editing LLMs. I would also like to see its generalization performance not only on `VQA-Constraints`, but also on other standard VLM benchmarks, such as MMMU.
- While the authors finding clearly indicates VLMs store and transfer information differently than text-only LLMs, I'm hoping that the authors could give a more detailed explanation to the cause of this. Since all tested VLMs are fine-tuned from LLMs, it is conceivable that they would operate in a similar fashion.
- This work currently only examines multimodal fusion at the embedding level, while there are other popular approaches such as the Flamingo architecture. I'm curious to see whether the authors finding would still hold under these different architectural choices.

**Questions:**

See weaknesses.

**Limitations:**

The authors have adequately addressed limitations and potential societal impacts of their work

---

> ### Author Rebuttal · Authors · 2024-08-06
>
> We thank the reviewer for providing constructive feedback and appreciating the paper’s presentation, techniques and the findings. We are glad that the reviewer feels that VQA-Constraint is a valuable contribution to the research community.
>
> Below we address the weaknesses raised by the reviewer:
>
> **While MultEdit has demonstrated impressive editing efficiency and generalization performance, the technical novelty is somewhat limited as it is an application of a well-tested technique for editing LLMs**: While the high-level idea of employing a closed-form approximation to update (edit) a targeted localized weight matrix in the network is similar to LLMs, our proposed editing method has two key distinctions:  (i) our editing method does not require a sample of Wikipedia text to be cached. In contrast, in LLM editing techniques like ROME [23], this is required for computing the uncentered covariance to ensure that the values corresponding to keys for unrelated texts remain similar (see Eq (2)). In our case, we enforce this condition with an L2 regularizer which ensures that the updated weight matrices do not deviate significantly from the original weight matrices (controlled by a hyperparameter \lambda - see Eq (5) in our paper). We find that this simple modification leads to strong editing results. One could use ROME to update MLLMs, but that would require caching the embeddings from a multimodal Wikipedia type dataset (which might not be readily available and clean for our use-case) thus adding an additional operation.   (ii) Our method *only* optimizes the multimodal language modeling next-token prediction loss. In contrast, LLM editing techniques optimize the language modeling loss along with a KL divergence loss which preserves the essence of the subject. In fact, during our early experimentations in the project, we used an additional KL divergence term to preserve the essence of the visual constraint. In particular, the objective of the KL divergence loss was to maintain the output probability distribution with the prompt (visual prompt (from visual tokens) + <visual-constraint> is a ) between the original MLLM and the MLLM whose value vector is optimized. However, empirically we did not see an improvement in the editing performance. Therefore we use the value vectors obtained with just the multimodal language modeling next-token prediction loss.
>
> Overall, MultEdit is simpler to implement and does not require caching a multimodal Wikipedia entry while leading to strong editing performances. We will add these distinctions in the final version of our paper.
>
> We note that MultEdit is one of our paper’s contributions. The others include MultimodalCausalTrace, a multi-modal casual tracing methodology, and the dataset VQA-Constraints. Together these contributions advance our understanding of how large multimodal language models process information. Each required technical novelty in terms of handling multimodal information and architectures, besides adapting current techniques.
>
> **I would also like to see its generalization performance not only on VQA-Constraints, but also on other standard VLM benchmarks, such as MMMU**: Thank you for the suggestion! We have evaluated the edited LLaVa on MMMU and report the following results:  LLaVa-7B (unedited) obtains 34.4% on the MMMU validation set, and LLaVa-7B (averaged across multiple edits) obtains 33.8% on the MMMU validation set. This result highlights that targeted model editing does not impact generalization performance significantly. We use the evaluation scripts from https://github.com/BAAI-DCAI/Bunny for the evaluation.
>
> **While the authors finding clearly indicates VLMs store and transfer information differently than text-only LLMs, I'm hoping that the authors could give a more detailed explanation to the cause of this. Since all tested VLMs are fine-tuned from LLMs, ....**: The reviewer raises a good point. Through conducting our analyses, we developed several hypotheses as to why MLLMs retrieve information differently compared to LLMs. We suspect that the language model elicits distinct internal circuits (a set of MLPs and attention heads) in the presence and absence of visual information. If the visual information is not present, the language model relies on one type of circuit, but in the presence of information from visual tokens, this circuit gets overridden by another.  It can also be crucial to study what happens to the visual tokens, after the projection stage (i.e., what type of information flows into the final architecture and how are the constraints encoded) as that is the potential orchestrator towards eliciting a different circuit in the model. Our work lays down a strong foundation and a practical set of tools (that we plan to open source) to study these hypotheses. We believe that making these results and tools available will also help us initiate a discussion and welcome hypotheses proposals from the community that can be studied in future work.
>
> **This work currently only examines multimodal fusion at the embedding level, while there are other popular approaches such as the Flamingo architecture. I'm curious to see whether the authors finding would still hold under these different architectural choices**: We scoped our paper to focus on the embedding-level fusion MLLM family (e.g., LLaVa, Multimodal-Phi3) because it has generally demonstrated stronger performance across a wide range of multi-modal tasks, compared to other families. Models like Flamingo fall into another MLLM family where visual tokens are fused with the language decoder at different layers. Because of these mechanistic differences, we expect that the causal layers will be different, however we leave this to future work to confirm. We note that since our framework and methodologies (including the probe datasets) can be applied to *any* MLLM architecture, they could be used to conduct these future analyses.

---

> > ### Comment · Reviewer_ieTM · 2024-08-12
> >
> > Thank you for your detailed response! I have increased my score accordingly.

---

### Author Rebuttal · Authors · 2024-08-06

We thank all the reviewers for their constructive feedback and comments. We have individually addressed the comments in their respective sections.

We want to highlight that our paper proposes a package of novel contributions that work together to advance our understanding of how state-of-the-art multimodal models process visual and textual information. Our proposed methodologies, MultiModalCausalTrace and MultEdit are carefully designed, extending existing approaches for causal tracing and model editing to multimodal inputs and are generalizable across multiple MLLM families. Together with our proposed dataset, VQA-Contraints, we present a wide suite of novel insights on how MLLMs store/retrieve information, and how information can be inserted into them in a computationally efficient way. We will open source all our methods, datasets and code to the community to enable the community to drive further advances in MLLM interpretability.

---

### Public Comment · ~Moulik_Choraria1 · 2025-04-15
**Code and Dataset Availability**

Has the VQA Constraints dataset been made public? Given that it is listed as one of the key contributions of this paper and as well as claimed in the abstract, I find it very surprising that there are no obvious links to the dataset in the exposition or any clear visibility online four months after the main conference.

EDIT: Thanks to the authors for sharing. Before this, I was unable to find the code either via standard searches or contacting the corresponding author. Therefore, it might be better to add a direct link to the paper/arXiv to increase visibility.

---

> ### Public Comment · ~Samyadeep_Basu1 · 2025-04-15
> **Code and dataset**
>
> Our code and data is public: https://github.com/microsoft/MLLMInterpret .

---

### Decision · Program_Chairs · 2024-09-25

**Decision:**

Accept (poster)

**Comment:**

The submission received scores of Accept, Accept, Weak Accept and Borderline Accept.

The paper applies and adapts interpretability techniques known for LLMs for Multimodal LLMs. The paper is thorough in providing insights about how MLLMs encode information but also how to leverage this knowledge to adapt them to new concepts. The reviewers are mostly positive about the work but point out that the contribution is limited as it mostly adapts techniques know for LLMs. However overall still recognize the value of adapting these techniques for LLMs and the insights derived from this exercise.